# Facile Synthesis of Metal-Impregnated Sugarcane-Derived Catalytic Biochar for Ozone Removal at Ambient Temperature

Reginald A. Verdida [1,2] , Alvin R. Caparanga [1] and Chang-Tang Chang [3,*]

1 School of Chemical, Biological and Materials Engineering and Sciences, Mapua University, Manila 1002, Philippines
2 School of Graduate Studies, Mapua University, Intramuros, Manila 1002, Philippines
3 Department of Environmental Engineering, National I-Lan University, Yilan 260, Taiwan
* Correspondence: ctchang@niu.edu.tw; Tel.: +886-939034143

**Abstract:** This study presents the first attempt at employing catalytic biochar to remove ground-level ozone at ambient temperature. With the increase in human activity, ozone has become a critical inorganic pollutant that needs to be addressed, using more sustainable methods. Fe- and Mn-impregnated catalytic biochars were prepared from a sugarcane feedstock via the wet impregnation method and pyrolysis at various temperatures, where the optimum value was determined to be 550 °C. The metal-impregnated biochar samples demonstrated enhanced surface areas and pore volumes compared with the pristine biochar (SCB550), resulting in improved ozone-adsorption capacity. SCB550-Fe exhibited an ozone-adsorption capacity of 52.1 mg/g at 20 ppm, which was approximately four times higher than that of SCB550. SCB550-Fe demonstrated superior ozone-removal performance compared to SCB550-Mn; 122 mg/g capacity as opposed to 116.2 mg/g at 80 ppm, respectively. Isothermal and kinetic modeling are also presented to suggest a plausible mechanism of ozone removal by catalytic biochar. This includes physical adsorption, complexation, electrostatic interaction, and electron transfer during the redox reaction between ozone and metals. Overall, this study should provide preliminary insights into ozone removal using biochar and promote further research regarding material optimization and kinetic studies.

**Keywords:** catalytic biochar; metal impregnation; ozone removal; sustainable





## 1. Introduction

Ozone, an allotrope of oxygen, is mostly thought of as the stratospheric barrier to ultraviolet radiation and is regarded as a powerful oxidant for various applications in different industry sectors. However, ozone is also considered one of the most critical inorganic pollutants due to its high reactivity and adverse health effects [1]. Indoor sources of ozone include photocopiers, laser printers, disinfection equipment, and air purification technologies such as ion air purifiers [2]. It can also be initiated by complex photochemical reactions between different pollutants such as nitrogen oxides (NOx), sulfur oxides (SOx), carbon monoxide (CO), volatile organic compounds (VOCs), and sunlight [3]. Additionally, extensive use of ozone in various applications, e.g., disinfection for water treatment [4], animal husbandry [5], color abatement for textiles [6], and food and beverage [7], results in direct ozone emission. Consequently, the USA Occupational Safety and Health Administration (OSHA) has set Public Health Air Standards of 0.1 ppm for 8 h and 0.3 ppm for 15 min as the threshold amount of ozone to which people can be safely exposed [8]. Long-term exposure to higher concentrations of ozone can pose serious health complications such as asthma [9], emphysema, and chronic bronchitis [2]. With this regard, it is fundamentally important to develop materials and methods of ozone removal for public health and environmental protection.

The most common approaches used for ozone removal include photocatalytic degradation [3], catalytic decomposition over noble metals [10–12] and transition metal oxides [13–15], and activated carbon adsorption [16–18]. Among these approaches, catalytic ozone decomposition using MnOx catalysts is the most widely adapted ozone removal method owing to its high efficiency even at long hours of exposure [14]. However, catalysts' performance on ozone removal has been consistently reported to plummet down at high relative humidity conditions due to the competitive adsorption of aqueous vapor to the active sites of the catalyst [19]. Although modifications such as co-precipitation [12], metal-organic framework [20], and supports [21] can improve catalyst resistance to harsh conditions, the preparation process is generally complex and expensive. Alternatively, another compelling option for ozone removal is utilizing activated carbon (AC) as the sorbent material for ozone adsorption. Namradi et al. compared the ozone removal capabilities of nine different AC materials from different sources, including coal-based, coconut shell-based, and palm shell-based AC, with various modifications [22]. They found that all AC samples have good ozone removal performance up until 200 ppm of ozone, specifically coconut-shell-based AC, which maintained ~95% efficiency for 6 h, and a further increase in ozone concentration rapidly deteriorates all ACs' efficiency. Furthermore, it is important to note that studies on the AC adsorption of ozone have failed to elucidate the probable influence of relative humidity on the efficiency of the materials.

Biochar, just like AC, is a carbonaceous material widely used for various applications such as soil quality remediation, wastewater treatment, and the adsorption of other emerging pollutants [23]. Being a low-cost sustainable material, it has aroused considerable interest in the research community due to its unique characteristics, such as abundant functional groups, tunable surface area, and well-defined pore structure [24]. Besides carbon, biochar also contains many other element species that affect and determine its corresponding properties and functions, such as H, O, S, and trace amounts of metals [25]. Furthermore, biochar acts as a bank of electron receivers and donors, as well as having buffering abilities between acids and bases [26]. However, characteristics of as-prepared pristine biochar, such as its low surface area and poor porosity, hamper its catalytic performance in various applications. For this reason, many studies have focused on understanding and optimizing biochar preparation, including the effect of biomass precursors, pyrolysis conditions, and the type of modification [27].

In order to obtain the desired surface and textural properties of biochar, it is necessary to understand the chemical composition of the biomass precursor being used. A vast variety of biomass is available, such as woody and non-woody biomass, industrial waste biomass, agricultural biomass wastes, and other lignocellulosic biomass that can be considered for biochar preparation [28]. The components in lignocellulosic biomass such as hemicellulose, cellulose, and lignin have diverse decomposition temperatures and degradation rates [29]. It has been reported that feedstocks with higher lignin content tend to have higher char yield compared to those with higher hemicellulose content [30]. However, the surface structure and porosity of the char derived from cellulose are superior compared to those of lignin [31]. Similarly, feedstocks with a higher fixed carbon content will produce a higher char yield compared to those with a higher volatile content after thermal treatment [32]. Finally, some studies also show that biomass with a lower ash content has higher porosity, and a higher ash content causes a lower surface area as ash blocks the pores of biochar, which ultimately limits its performance in most of its applications [33].

In addition, pyrolysis conditions also have a significant influence on the distribution of pyrolysis products. Slow pyrolysis favors high biochar yields, while the fast pyrolysis process is better suited for maximizing bio-oil formation [34]. Furthermore, the rise in pyrolysis temperature could contribute to the decrease in biochar yield due to the fact that higher temperatures can trigger the secondary cracking reaction of pyrolytic vapor, thereby yielding more gaseous products [35]. Additionally, the heating rate is another important process parameter that has a remarkable effect on the composition and properties of biochar. A higher heating rate contributes to the further thermal decomposition of lignocellulosic

biomass since the limitations of heat and mass transfer are alleviated, forming a more defined pore structure [36]. Taking these parameters into consideration, low temperature, low heating rate, and long retention times are favorable to the formation of biochar [37]. In general, heat transfer from the pyrolysis reactor to the biomass precursor is a key factor affecting biomass pyrolysis reactions and product distribution.

However, the relatively limited adsorption capacity and performance of as-prepared pristine biochar on various applications still cannot warrant its viability for practical applications. Recent studies have reported that transition metals introduced into carbon-based biomass precursors such as Fe and Mn can augment the physicochemical properties of pristine biochar and lead to better catalytic performance [38]. Aside from specific surface area, it can also effectively modify the electronic structure of carbon-based materials, such as by generating more active sites [39], improving electron transfer ability [40,41], and conferring magnetic properties [42], which are all conducive to enhancing their adsorption capacity. Several studies also discovered relatively the same enhancement in biochar characteristics and performance upon doping with bimetallic FeMn metal salts [43]. However, there have not been any studies regarding the application of Fe or Mn metal salts impregnated on biochar specifically for catalytic ozone decomposition in the gaseous phase. Herein, Fe and Mn-functionalized catalytic biochar were prepared for catalytic ozone decomposition in the gaseous phase at ambient temperature.

To the best of our knowledge, the use of biochar as the precursor carbonaceous material for ozone removal has never been reported. In this work, the potential of catalytic biochar derived from sugarcane bagasse as a catalyst for ozone removal has been investigated. Biochar was modified using the facile incipient wet impregnation method of Fe and Mn metal salts to enhance surface morphology and chemistry, as substantiated through different characterization techniques. Pristine and catalytic biochar are assessed for ozone removal efficiency in terms of varying pyrolysis temperatures and ozone concentrations. Finally, adsorption kinetics and isothermal models were proposed in order to describe the catalytic decomposition mechanism of ozone on the surface of the biochar.

## 2. Results and Discussion

### 2.1. Determining Optimum Pyrolysis Temperature

The physicochemical characterizations of sugarcane biochar (SCB) pyrolyzed at different temperatures were elucidated to determine the ideal pyrolysis temperature meant for modification. Table 1 shows that biochar yields considerably decreased as the temperature increased from 350 to 550 °C (31.61% to 23.39%, respectively) and relatively stabilized as the temperature was further increased to 650 °C (22.09%). This suggests that the main carbonization process has been completed at around 550 °C and most of the volatile organic matter has been decomposed [44]. Additionally, hydrogen (H) and oxygen (O) content both decreased with the rise in pyrolysis temperature, which is a result of dehydration and decarboxylation reactions of the weak linkages within the SCB structure during thermal cracking [45]. Conversely, the carbon (C) content of biochar was positively correlated with the pyrolysis temperature, suggesting a continuing condensation of carbon into aromatic structures [46]. Moreover, it can be observed that there is a sudden increase in nitrogen fractions in the biochar produced at 650 °C (from 0.98% to 1.14%), which may be due to the association of nitrogen with complex structures in the biochar that did not undergo alteration or decomposition at the said temperature [47]. Furthermore, aside from carbon, substances that have not been volatilized may include trace amounts of inorganic (Ca, Na, Mg, K, Al, and Si) and lignin/cellulose components from the biomass itself [48].

**Table 1.** Elemental analysis of sugarcane biomass pyrolyzed at various temperatures.

| Sample | Biochar Yield (%) | C (%) | H (%) | O (%) | N (%) | O/C | H/C |
|---|---|---|---|---|---|---|---|
| SCB350 | 31.61 | 72.30 | 3.47 | 17.08 | 1.07 | 0.23 | 0.05 |
| SCB450 | 27.04 | 81.13 | 2.83 | 11.25 | 1.15 | 0.14 | 0.03 |
| SCB550 | 23.39 | 84.45 | 1.96 | 9.82 | 0.98 | 0.12 | 0.03 |
| SCB650 | 22.09 | 86.73 | 1.42 | 9.30 | 1.04 | 0.11 | 0.02 |

Meanwhile, elemental ratios of H/C and O/C decline. The H/C ratio is a conventional metric used for assessing the degree of carbonization, hydrophobicity, and aromaticity [49]. On the other hand, O/C ratios can reflect the polarity, hydrophilicity, and content of oxygen-containing functional groups [50]. The decrease of H/C and O/C ratios suggests that during high-temperature thermal treatment, the carbonization of biochar and degree of aromatization were enhanced; however, the polarity and oxygen functional groups were reduced. These results are consistent with existing literature on sugarcane bagasse pyrolysis [51]. Although an increase in the polarity of biochar samples is desired for better ozone removal due to enhanced electron transfer, the presence of surface functional groups, most notably oxygen-containing ones, is also deemed beneficial for ozone decomposition. A balance between the existence of both properties should be taken into great consideration when choosing the optimum pyrolysis temperature.

Surface functional groups of biochar produced at different temperatures were further elaborated using FTIR analysis (Figure 1). Five characteristic bands were observed at 3370, 2890, 1690, 1560, and 1030 $cm^{-1}$. The presence of the band at 3370 $cm^{-1}$ is ascribed to the stretching vibration of -OH, which was clearly detected only at samples SCB350 and SCB450, suggesting probable dehydration of phenolic groups as the temperature in pyrolysis was increased [52]. The same observation can be identified on the band located at 2890 $cm^{-1}$ that corresponds to asymmetric and symmetric axial deformation of aliphatic C–H [47]. The intensity region at 1690 and 1560 $cm^{-1}$ is correlated to the stretching vibration of aromatic C=O and C=C, which has been reduced for the SCB650 biochar, indicating that this temperature was sufficient to break the bonds corresponding to the alkene and carbonyl groups [53]. All samples showed the band at 1030 $cm^{-1}$, characteristic of C-O stretching, and close to 1160 $cm^{-1}$ for the asymmetrical stretching of C-O-C [54]. The observed absorbance bands in biochar samples are typical for biomasses consisting of cellulose, hemicellulose, and lignin [55,56]. In general, the effect of increasing pyrolysis temperature on the FTIR profiles of biochar decreased the amount of surface functional groups, particularly phenolic, carbonyl, and carboxylic groups. Some regions found in samples pyrolyzed at lower temperatures were absent in SCB650, indicating that increased energy in the system caused disruption, breaking bond characteristics in those particular regions. Previous studies have reported the same outcomes using various biomass precursors [57].

The surface morphology of the biochar produced at various pyrolysis temperatures is characterized using SEM (Figure 2). The micrographs of the biochar pyrolyzed at low temperatures (SCB350 and SCB450) showed that the main plant structure was preserved after the thermal treatment of the biomass and marginal changes in the textures of the surfaces of the materials produced [58]. Moreover, the development of a more defined pore structure was seen with increasing pyrolysis temperature. This phenomenon is due to the fact that higher temperatures can trigger the secondary cracking reaction of organic compounds from the feedstock, thereby yielding more gaseous products [59]. The escape of the volatile compounds on the surface of the biochar generated cracks and pores, which translates to a higher surface area which is deemed beneficial for the removal of ozone [60]. However, a further increase in temperature could lead to progressive decomposition of the volatile organic compounds, thereby enhancing the yield of on-condensable gases and collapsing the walls of the pores [61].

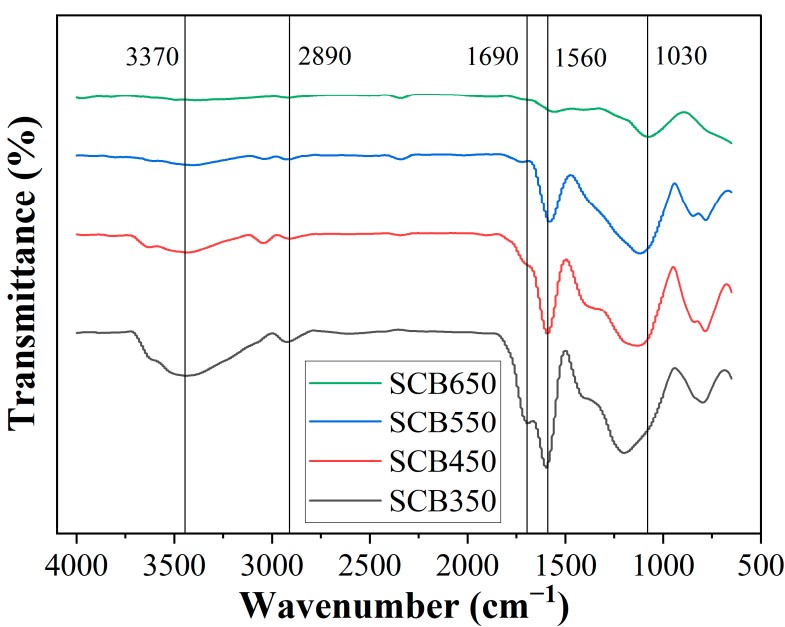

**Figure 1.** FTIR spectra of sugarcane biomass pyrolyzed at various temperatures.

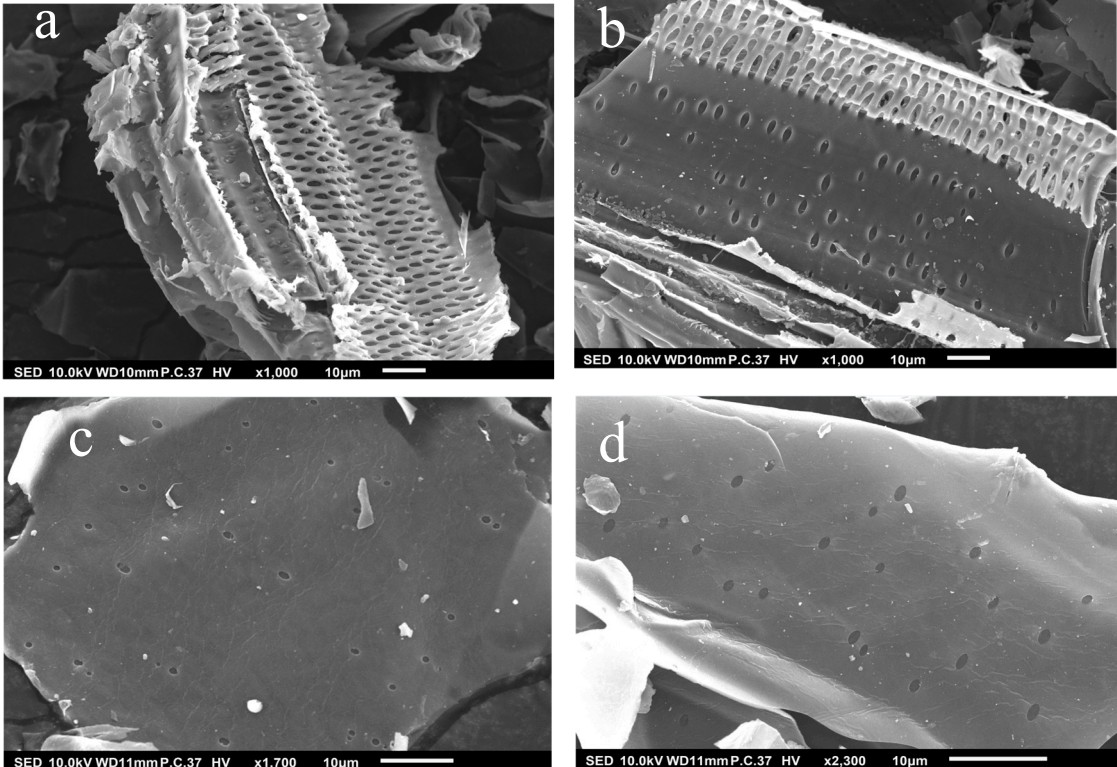

**Figure 2.** SEM images of (**a**) SCB350, (**b**) SCB450, (**c**) SCB550, and (**d**) SCB650.

Interestingly, considering 10 representative pores from the SEM images shown in Figure S1, researchers calculated a decrease in the average pore size of samples pyrolyzed at 350, 450, and 550 °C (1.58, 1.35, and 0.82 μm, respectively). However, when the pyrolysis temperature was further increased to 650 °C, the average pore size increased to 0.91 μm. This could be attributed to the fact that with increasing temperature, more burn-off occurred, which developed a broad range of pore networks [62]. However, more decomposition takes place at higher temperatures, which might have widened the micropores to mesopores or macropores because of the etching effect and ultimately decreased the surface area of the

biochar [63]. Further volatilization of organic matter in the internal structure of the biochar with progressive thermal treatment leads to the formation of deep channels with clear pores that appear more prominently with increasing temperature [64]. This behavior could be confirmed from previous studies where a decrease in average pore width from 1.93 to 1.42 nm was observed with increasing temperature from 600 to 700 °C and an increase to 1.72 nm at 800 °C [65].

By combining all the information regarding biochar yield, degree of aromaticity and polarity, amount of surface functional groups, and surface morphology of SCB pyrolyzed at different temperatures, it can be deduced that sample SCB550 is the optimum pyrolysis temperature for modification. SCB550 portrayed the smallest average pore diameter of 0.82 nm, considering 10 representative pores, which can boost the physical adsorption of ozone in addition to containing an ample amount of surface functional groups. Oxygen-containing functional groups can promote the complexation or decomposition of ozone on the surface of the biochar during adsorption. Furthermore, the existence of polarity and good electron transfer ability of the sample can also help with the removal of ozone through electrostatic interaction at the surface of the biochar. Additionally, the performance of biochar pyrolyzed at 350, 450, 550, and 650 °C on the removal of ozone was evaluated at 15 ppm. It was found that among the biochar samples, the highest adsorption capacity recorded belongs to biochar pyrolyzed at 550 °C at 8.23 mg/g. The adsorption capacities of all samples for this test are presented in Figure S2. Hereinafter, SCB550 was designated as representative pristine biochar for modification, characterization, ozone removal tests, and isothermal and kinetic modeling.

### 2.2. Characterization of Modified Biochar

Metal salt impregnation causes biochar to undergo substantial transitions in terms of physical-chemical reactions as the pyrolysis process progresses. These inorganic compounds interact within the internal structure of the biomass during the pyrolysis process and modify the resulting properties of the biochar [66]. Previous studies attempted to illustrate plausible transformations in organic phases of biochar involving several chemical reactions occurring throughout the thermal treatment process, including dehydration, polymerization, decarboxylation, dichlorination, crystallization, and reduction-like reactions [67]. These transitions during the pyrolysis process, brought about by the impregnation of metal salts, cause substantial changes in the physicochemical characteristics of catalytic biochar.

The FTIR spectra of catalytic biochars are presented in Figure 3 to distinguish changes in surface functional groups after modification. All characteristic bands of as-prepared pristine biochar are still observed after impregnation of metal salts; however, they are relatively decreased, confirming further gasification and conversion to the graphitic structure brought about by metal impregnation. Previous studies report that secondary cracking reactions of organic compounds in the biomass were enhanced with the addition of inorganic salts prior to pyrolysis [68]. Moreover, the presence of inorganic salt could enhance the carbonization reactions in biomass pyrolysis, thereby yielding fewer surface functional groups [37]. Additionally, the bands at 1030 and 1560 $cm^{-1}$, which correspond to C-O and C=C stretching, respectively, are significantly reduced on SCB550-Fe compared to SCB550-Mn, suggesting a more active catalytic activity of Fe in the thermal cracking of the organic compounds in the biomass during pyrolysis.

The surface morphology of prepared catalytic biochar showed an enhanced heterogeneous and porous structure (Figure 4). Compared to pristine biochar SCB550, the resulting catalytic biochar materials develop nanoscale pores and more obvious surface defects that are beneficial for better ozone removal performance. It has been reported that some metal salts (e.g., $AlCl_3$, $FeCl_2$, and $MnCl_3$) exhibit catalytic activity during the pyrolysis process, increasing porosity and creating a more positively charged surface [69]. The pyrolysis process can support the development of pores as it dehydrates the char and opens pores through cellulose degradation reactions [70].

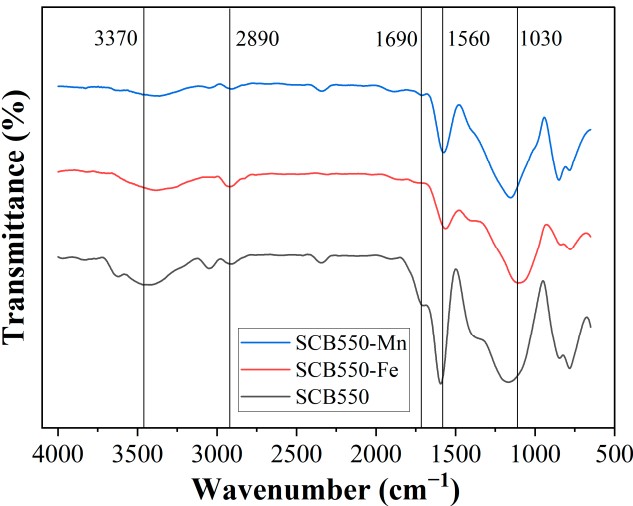

**Figure 3.** FTIR spectra of catalytic biochar SCB550-Fe and SCB550-Mn in contrast with pristine biochar SCB550.

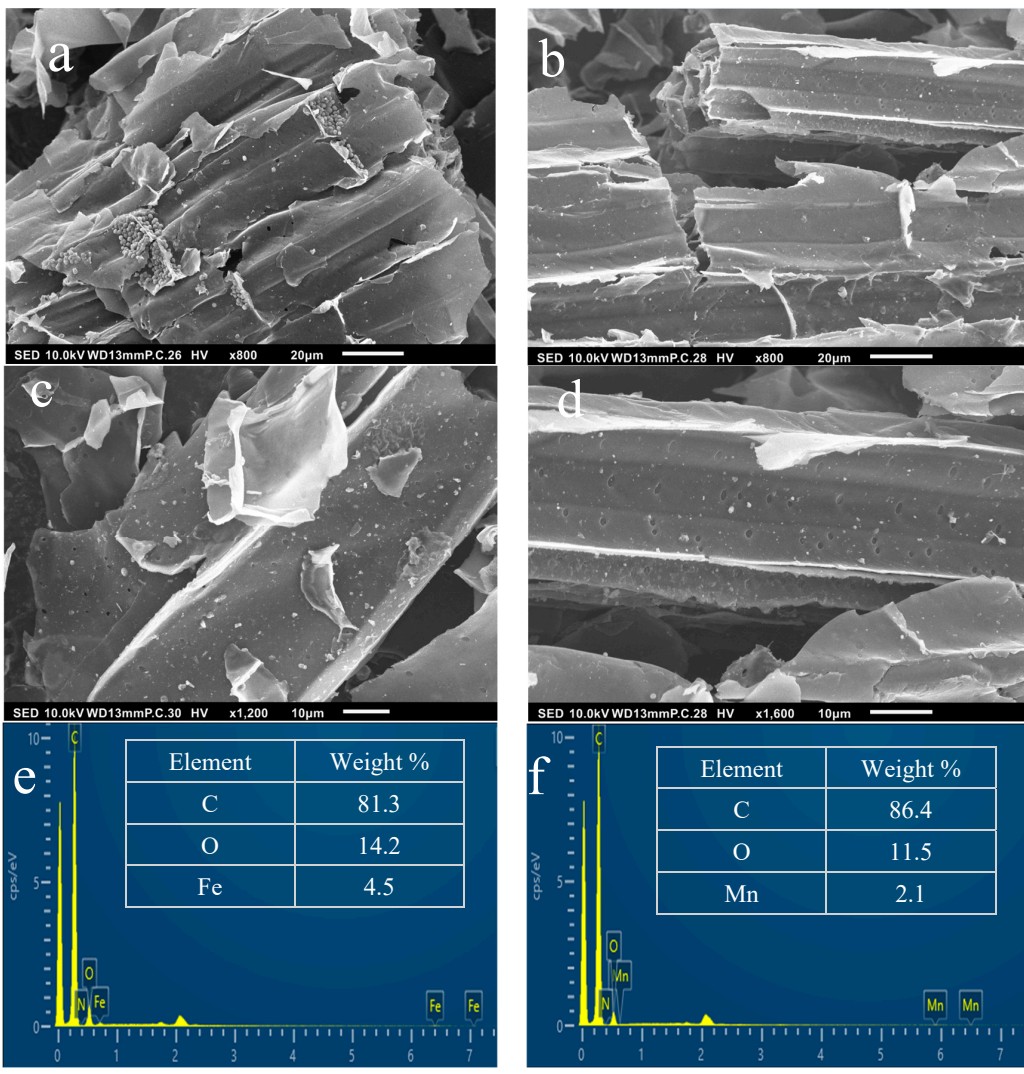

**Figure 4.** SEM images of catalytic biochar (**a**,**c**) SCB550-Fe and (**b**,**d**) SCB550-Mn at 800× and 1600× magnifications. EDS analysis of (**e**) SCB550-Fe and (**f**) SCB550-Mn.

Furthermore, SEM images also show crystalline particles of Fe and Mn salts scattered throughout the surface of the biochar. These crystals prove the successive impregnation of metals on the surface of biochar. Successive impregnation of metal salts was proven by the EDS analysis presented in Figure 4e,f. From the EDS analysis, it can be seen that Cl is not present in the samples, which were impregnated with $FeCl_3$ and $MnCl_2$. This means that Fe and Mn completely transformed into their oxide forms during the impregnation and pyrolysis processes. These results are consistent with the work of Tomin et al., who tailored 18 different types of metal-impregnated biochar [71]. Moreover, the EDS elemental mapping images presented in Figure S3 showed no particular bright spots in Fe and Mn scans, indicating that both metals are evenly dispersed on the surface of the biochar.

Table 2 summarizes the BET surface area, pore volume, and average pore size of the pristine and catalytic biochar produced. SCB550-Fe exhibited the largest specific surface area (371.1 $m^2$/g) and pore volume (0.159 $cm^3$/g) among the samples. It can also be deduced that both catalytic biochar demonstrated a significant increase in surface area and pore volume compared to pristine biochar (299.6 $m^2$/g). This was due to the dissolution of organic components by metal salts in the impregnation stage and facilitated the chemical reaction in the process of pyrolysis, which resulted in the formation of a more porous structure [72]. There is also a significant increase in the pore volume of SCB550 (0.112 $cm^3$/g) when compared to SCB550-Fe (0.159 $cm^3$/g) and SCB550-Mn (0.158 $cm^3$/g). The slight difference between the surface area and pore volume of SCB550-Fe (371.1 $m^2$/g and 0.159 $cm^3$/g, respectively) and SCB550-Mn (345.5 $m^2$/g and 0.158 $cm^3$/g, respectively) may be caused by the relative catalytic activity of particular metal salts during impregnation and pyrolysis. Surface area and porosity play an important role in ozone removal. The enhanced surface area and porous structure of the catalytic biochars act as a host for the metal particles, which can further contribute to the decomposition of ozone through electron transfer. Additionally, it can further promote physical adsorption through surface precipitation and the complexation of ozone.

**Table 2.** BET surface area, pore volume, and pore size of pristine and catalytic biochars.

| Biochar | $S_{BET}$ ($m^2$/g) | $S_{micro}$ ($m^2$/g) | $V_{total}$ ($cm^3$/g) | $V_{micro}$ ($cm^3$/g) | $d_{ave}$ (nm) |
|---|---|---|---|---|---|
| SCB550 | 299.6 | 264.5 | 0.112 | 0.123 | 1.49 |
| SCB550-Mn | 345.5 | 293.7 | 0.158 | 0.136 | 1.91 |
| SCB550-Fe | 371.1 | 318.4 | 0.159 | 0.148 | 1.76 |

Analogously, it can be observed that there is an increase in the average pore diameter of SCB550 (1.49 nm) compared to SCB550-Fe (1.76 nm) and SCB550-Mn (1.91 nm). This observation might be due to the catalytic influence of Fe and Mn salts during the pyrolysis process. These results are consistent with previous studies impregnating biochar with several metal salts/oxides [73]. Furthermore, between the two catalytic biochars, SCB550-Fe exhibited a larger micropore area (318.4 $m^2$/g) and volume (0.148 $cm^3$/g), as well as a smaller average pore width (1.76 nm) compared to SCB550-Mn (293.7 $m^2$/g, 0.136 $cm^3$/g, and 1.91 nm, respectively). These properties are beneficial for better ozone removal performance due to the availability of more active sites. Similar results were also reported in other previous studies and are summarized in Table S1.

### 2.3. Ozone Breakthrough Experiments

Figure 5 shows the ozone breakthrough curves of both pristine and catalytic biochar under a relative humidity of 35% and 0.47 retention time at room temperature. SCB550 were tested with a lower range of ozone concentrations (5, 10, 15, and 20 ppm), while SCB550-Fe and SCB550-Mn were assessed on a much higher range of ozone (20, 40, 60, and 80 ppm) due to their significantly enhanced ozone removal performance. At 20 ppm ozone level, SCB550 immediately reached saturation at 246 min (Figure 5a), while SCB550-Fe and SCB550-Mn saturated at 675 and 745 min, respectively (Figure 5b,c). These results imply an approximately four-fold enhancement in the ozone removal performance of catalytic

biochar from SCB550, which may be due to the availability of more active sites brought about by larger specific areas and impregnated metal salts. Thus, catalytic biochar was assessed with higher concentrations of ozone to better elucidate its removal performance. Meanwhile, it can also be observed from the breakthrough curves that an increase in ozone concentration translates to a decrease in breakthrough ($C/C_0 = 0.1$) and saturation times ($C/C_0 = 0.9$). This may be due to the fact that higher concentrations of ozone in the bulk fluid can more rapidly occupy active sites on the surface of the biochar.

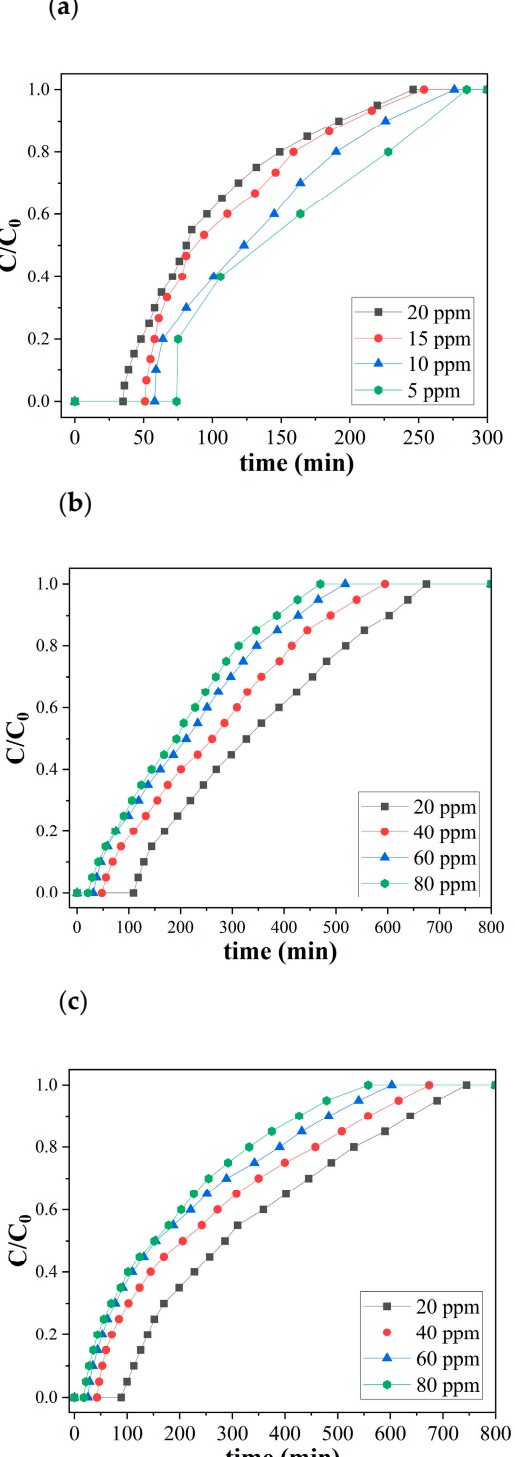

**Figure 5.** Breakthrough curves of (**a**) SCB500, (**b**) SCB550-Fe, and (**c**) SCB550-Mn (RH = 35%, temperature = 23 °C, retention time = 0.47 s, and packing density = 0.03 g/cm$^3$).

Moreover, comparing the two catalytic biochars, it can be observed that in the first 180 min of adsorption, SCB550-Fe and SCB550-Mn showed rapid adsorption of ozone, with removal efficiencies plummeting down to 78% and 68% (20 ppm). It is possibly due to the deactivation of numerous adsorption sites and the driving force of the ozone concentration gradient in the bulk fluid to the surface of the biochar. This also suggests that SCB550-Fe has superior ozone removal performance compared to SCB550-Mn. After 180 min, the amount of ozone adsorbed by both catalytic biochars seemed to stabilize until it reached equilibrium. Interestingly, across all initial ozone concentrations, SCB550-Mn had a longer breakthrough curve when compared to SCB550-Fe. However, calculating their adsorption capacities (Figure 6), SCB550-Fe demonstrates a higher capacity across all concentrations, reaching 122.8 mg/g at 80 ppm compared to SCB550-Mn (116.2 mg/g). This indirect proportionality might be due to the faster deactivation of adsorption sites of SCB550-Mn in the first few minutes of the reaction with ozone (Figure 5c). Aside from the difference in ozone concentration between the bulk fluid and the biochar surface, adsorption capacity is also dependent upon the change in time (see Equation (7)). As previously noted, in the first few minutes where the change in ozone concentration is at its highest, SCB550-Mn rapidly saturates compared to SCB550-Fe, resulting in a lower overall adsorption capacity. Meanwhile, a comparison of the performance of catalytic biochar in this study with the literature is presented in Table 3. Though SCB550-Fe and SCB550-Mn demonstrated inferior removal efficiency than most literature, it is important to note that catalysts and other reported materials utilized for ozone removal are rather expensive, involve complex fabrication processes, and are often environmentally unfriendly. These materials lack the economical and sustainable benefits complemented by the facile synthesis method of prepared catalytic biochar presented in this study. Moreover, with the recent advancement in biochar research and development, further modification of biochar in future studies can enhance its affinity to ozone and compare favorably with other materials.

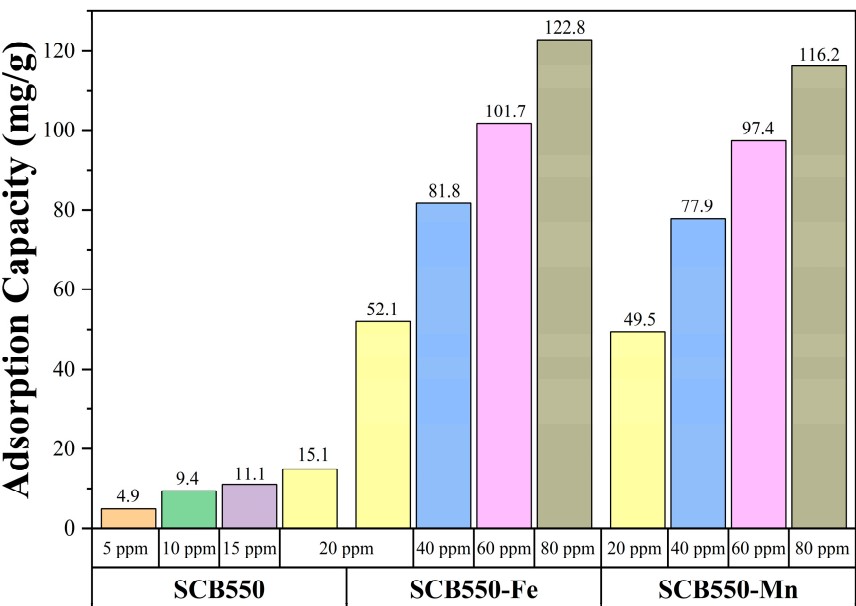

**Figure 6.** Adsorption capacities of pristine and catalytic biochars.

**Table 3.** 3-hour removal efficiency comparison of catalytic biochar with the literature.

| Sample Name | Type of Material | $O_3$ Concentration | Temperature & RH | 3-hour Removal Efficiency | Reference |
|---|---|---|---|---|---|
| ZZU-281 | Metal organic framework | 50 ppm | 25 °C, 40% | 100% | [20] |
| Ce-OMS-2 (95 °C, 24 h) | Catalyst | 40 ppm | 30 °C, 90% | 92% | [13] |
| MnO/AC-700 | Mn supported on activated carbon | 30 ppm | 25 °C, 45% | 94% | [21] |
| AC | Activated Carbon | 45 ppm | 25 °C, 60% | 51% | [74] |
| SCB550-Fe | Catalytic Biochar | 20 ppm | 23 °C, 35% | 78% | This work |
| SCB550-Fe | Catalytic Biochar | 40 ppm | 23 °C, 35% | 64% | This work |
| SCB550-Mn | Catalytic Biochar | 20 ppm | 23 °C, 35% | 68% | This work |
| SCB550-Mn | Catalytic Biochar | 40 ppm | 23 °C, 35% | 52% | This work |

*2.4. Kinetic Modelling*

The adsorption isotherms can provide important information about surface properties, affinity, and adsorption mechanisms. Langmuir and Freundlich's adsorption isotherm models were applied to evaluate the whole adsorption process (Figure S4). The isotherm parameters were summarized in Table 4. The Langmuir model had better fitting data and a higher correlation coefficient ($R^2$ = 0.991) for SCB550, which indicated that the mechanism of ozone adsorption can be interpreted as a monolayer adsorption process. On the other hand, the Freundlich model had a better fitting and higher correlation coefficients for SCB550-Mn (0.9990) and SCB550-Fe (0.9992), suggesting a multilayer mechanism of ozone adsorption. Additionally, the maximum adsorption capacity according to the Langmuir model can reach up to 200 mg/g and 204 mg/g for SCB550-Mn and SCB550-Fe, respectively. Furthermore, analysis using the Freundlich model shows that n values are increasing from 1.32 (SCB550) to 1.61 (SCB550-Mn) and 1.64 (SCB550-Fe), which indicates more favorable adsorption of catalytic biochar over SCB550 and superior performance of SCB550-Fe compared to SCB550-Mn.

**Table 4.** Summary of adsorption isotherm modeling parameters.

| | Langmuir | | | Freundlich | | |
|---|---|---|---|---|---|---|
| | $Q_{max}$ (mg/g) | $K_L$ | $R^2$ | n | $K_F$ | $R^2$ |
| SCB550 | 39 | 0.0232 | 0.9913 | 1.3170 | 1.5007 | 0.9812 |
| SCB550-Mn | 200 | 0.0162 | 0.9982 | 1.6119 | 7.6913 | 0.9992 |
| SCB550-Fe | 204 | 0.0170 | 0.9972 | 1.6388 | 8.4004 | 0.9990 |

The pseudo-first-order and pseudo-second-order were used to elaborate the kinetic mechanism of the ozone adsorption process (Figures S5–S7). The kinetic data were summarized in Table 5, which revealed that pseudo-second-order was a better fit than pseudo-first-order because of higher $R^2$ values across all concentrations on all biochar samples. Therefore, the theoretical adsorption capacity predicted by the pseudo-second-order model was closer to the experimental data. This implies that chemisorption was the main adsorption mechanism for both pristine and catalytic biochars. Therefore, the adsorption behavior of ozone may be due to the existence of a valence force through electron sharing between ozone and biochar, forming weak electrostatic interactions such as dipole-dipole and Van der Waals forces of attraction. Additionally, the rate constants of SCB550-Fe were the highest among the three biochars and relatively better than those of SCB550-Mn. This suggests the enhanced ozone removal capabilities of catalytic biochar compared to SCB550

(at 20 ppm) and the superior performance of SCB550-Fe compared to SCB550-Mn across all initial ozone concentrations.

**Table 5.** Summary of adsorption kinetics modeling parameters.

| | O₃ Concentration | Pseudo First-Order | | Pseudo Second-Order | |
|---|---|---|---|---|---|
| | | $K_1$ | $R^2$ | $K_2$ | $R^2$ |
| SCB550 | 5 ppm | 0.0009 | 0.9868 | 0.0352 | 0.9989 |
| | 10 ppm | 0.0008 | 0.9186 | 0.0591 | 0.9944 |
| | 15 ppm | 0.0004 | 0.9539 | 0.0619 | 0.9946 |
| | 20 ppm | 0.0002 | 0.9727 | 0.0563 | 0.9959 |
| SCB550-Mn | 20 ppm | 0.0005 | 0.7832 | 0.0773 | 0.9910 |
| | 40 ppm | 0.0004 | 0.8692 | 0.0924 | 0.9957 |
| | 60 ppm | 0.0003 | 0.8285 | 0.1044 | 0.9925 |
| | 80 ppm | 0.0001 | 0.8556 | 0.2069 | 0.9948 |
| SCB550-Fe | 20 ppm | 0.0009 | 0.8948 | 0.0873 | 0.9929 |
| | 40 ppm | 0.0005 | 0.8589 | 0.1185 | 0.9915 |
| | 60 ppm | 0.0004 | 0.8287 | 0.1524 | 0.9925 |
| | 80 ppm | 0.0001 | 0.8797 | 0.2801 | 0.9904 |

*2.5. Plausible Ozone Removal Mechanism*

Herein, we present the plausible removal mechanism of ozone in pristine and catalytic biochar. Primarily, the co-presence of both hydrophilic and hydrophobic sites in biochar has been proven to be beneficial in heterogenous catalysis involving substrates with varied polar functionalities, just like ozone [45]. The hydrophilic (polarity index) or hydrophobic property (aromaticity) of biochar highly depends upon the type and nature of existing functional groups on the surface. As a dipole molecule, ozone has both a nucleophilic site and an electrophilic site, which can react with the H (electrophilic) and O (nucleophilic) atoms on the surface of the biochar, developing electrostatic interactions. Furthermore, the surface of most pristine biochar is often negatively charged in association with its abundant oxygen-containing function groups [75], which can further promote dipole-dipole interaction with the partial positive side of ozone. Physical adsorption through bulk diffusion, film diffusion, and pore diffusion is proven through intraparticle diffusion modeling provided in Figure S8. Table 6 summarizes the intraparticle diffusion parameters calculated from the modeling, showing three adsorption stages.

**Table 6.** Intraparticle diffusion modeling parameters of SCB550-Fe.

| Concentration (ppm) | First Stage | | Second Stage | | Third Stage | |
|---|---|---|---|---|---|---|
| | $K_{bd}$ (mg g$^{-1}$ min$^{-0.5}$) | $r^2$ | $K_{fd}$ (mg g$^{-1}$ min$^{-0.5}$) | $r^2$ | $K_{id}$ (mg g$^{-1}$ min$^{-0.5}$) | $r^2$ |
| 20 | 0.0019 | 0.9754 | 0.0049 | 0.9932 | 0.0008 | 0.9841 |
| 40 | 0.0026 | 0.9853 | 0.0074 | 0.9951 | 0.0013 | 0.9836 |
| 60 | 0.0031 | 0.9754 | 0.0079 | 0.9902 | 0.0017 | 0.9899 |
| 80 | 0.0036 | 0.9965 | 0.0073 | 0.9963 | 0.0020 | 0.9909 |

Another plausible mechanism for the adsorption of ozone is through surface functional group complexation. The changes in the amount of surface functional groups can be observed in the comparison of the FTIR spectra of fresh and used SCB550-Fe shown in Figure 7. It can be observed that surface functional groups, particularly oxygen-containing groups, significantly decreased after the adsorption of ozone. Through chemisorption, ozone reacts with the biochar surface and forms oxygen-containing functional groups (Equation (1)) [22]. After which, further interaction with ozone produces oxygen, carbon monoxide, and carbon dioxide. Additionally, already existing oxygen-containing functional groups can react with ozone and generate the same gaseous products. Moreover, ozone can

also interact with surface -OH groups forming hydrogen superoxide and oxygen, which further contribute to the removal of ozone, as shown in Equation (8) [76].

$$C_n \overset{O_3}{\to} \begin{matrix} surface\ functional\ groups \\ (-COOH, -COH, -C=O) \end{matrix} + O_2 \overset{O_3}{\to} CO_2, CO + O_2 \tag{1}$$

$$OH^- + O_3 \to HO_2^- + O_2 \tag{2}$$

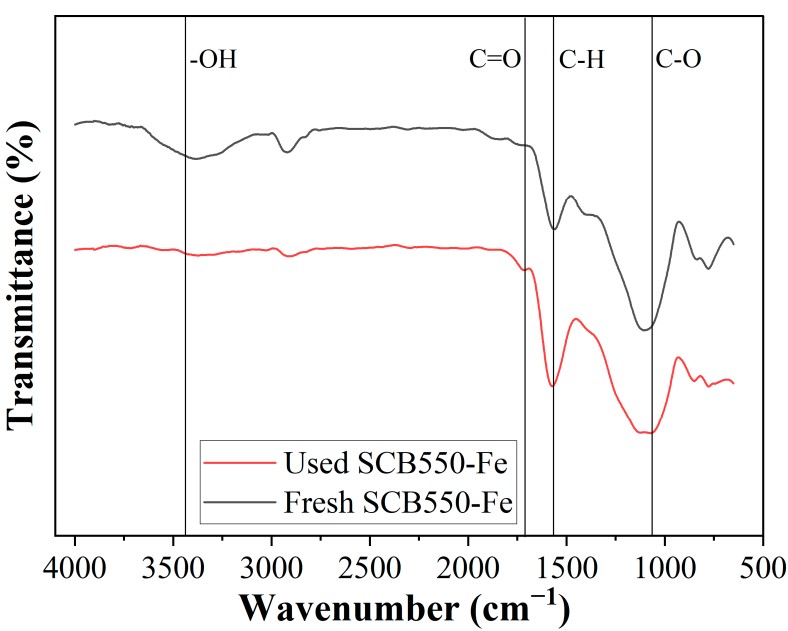

**Figure 7.** Comparison of the FTIR spectra of fresh and used SCB550-Fe.

Finally, by virtue of metal impregnation, Fe and Mn particles on the surface of the biochar are oxidized when made in contact with ozone through the transfer of electrons facilitating a redox reaction, as listed in Equations (3)–(5). Transition metals that appear in biomass can significantly affect the formation of redox-active active sites, acting as an electron acceptor for ozone decomposition during the adsorption process [77]. Basically, multivalent transition metals commonly have redox pairs (such as Co(II)/Co(III), Fe(II)/Fe(III), and Mn(III)/Mn(IV)). According to the literature, a low oxidation state is more desirable since $O_3$ can transfer electrons to them more rapidly thus facilitating ozone decomposition [78]. The slightly better performance of SCB550-Fe in ozone decomposition might be due to Fe (II)/Fe (III) lower reduction potential (~0.77 V) compared to Mn (III)/Mn (IV) of SCB550-Mn (~0.95V) since the transformation of Fe(II) to Fe(III) and Mn(III) to Mn(IV) directly correlates with its ozone decomposition capabilities [79].

$$O_3 + M^{n+} \to O_2 + O^- + M^{(n+1)+} \tag{3}$$

$$O_3 + O^- + M^{(n+1)+} \to O_2 + O_2^- + M^{(n+1)+} \tag{4}$$

$$O_2^- + M^{(n+1)+} \to O_2 + M^{n+} \tag{5}$$

Ultimately, the removal process of ozone by pristine and catalytic biochar involved multiple mechanisms (Figure 8), including physical adsorption, surface functional group complexation, and redox reactions. Firstly, owing to the large surface area and porosity of catalytic biochar, there were more active sites for the physical adsorption of ozone by electrostatic and dipole-dipole interactions. Moreover, ozone can react with the biochar surface to form oxygen-containing groups. Further interaction with ozone generates oxygen, carbon monoxide, and carbon dioxide. Finally, the presence of Fe and Mn on the surface of the biochar promotes ozone decomposition by redox reactions.

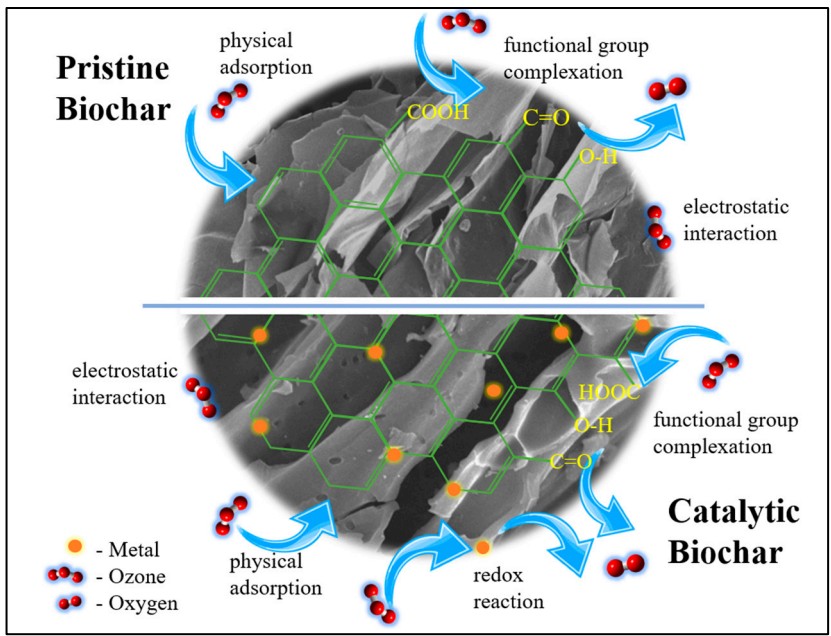

**Figure 8.** Plausible mechanism of ozone removal on the surface of pristine and catalytic biochar.

### 3. Materials and Methods

#### 3.1. Materials and Reagents

Raw sugarcane bagasse (SCB) was collected from Yilan County, Taiwan. The biomass was washed several times with ultrapure water to remove impurities and oven-dried at 110 °C for 24 h. Then the dried biomass was pulverized to pass through a 32-mesh sieve (0.5 mm), stored in a sealed, oxygen-free bag, and kept in a desiccator. Metal salt powders $FeCl_2 \cdot 4H_2O$ and $MnCl_2 \cdot 4H_2O$ were purchased from Nihon Shiyaku Reagents, which are of analytical grade and used without further purification. All working solutions were prepared using ultrapure water unless otherwise specified.

#### 3.2. Preparation of Pristine Biochar

Pristine biochar was prepared by slow pyrolysis. A certain amount of milled sugarcane bagasse is weighed and set in a crucible combustion boat. The crucible was then placed in a quartz tubular reactor and pyrolyzed in a tubular muffle furnace under $N_2$ at different temperatures of 350, 450, 550, and 650 °C. The rate of temperature ramp was at 5 °C·min$^{-1}$. The sample was heated from room temperature to the target temperature and maintained for 2 h in an $N_2$ atmosphere. Different temperatures were employed to investigate the effect on the surface chemistry and morphology of biochar. After cooling the chamber of the muffle furnace, the crucible was taken out, and the biochar was placed in a sealed, oxygen-free bag and kept in a desiccator. Samples were designated as SCB350, SCB450, SCB550, and SCB650, respectively.

#### 3.3. Preparation of Catalytic Biochar

The Fe- and Mn-modified catalytic biochars were prepared by the incipient wet impregnation method. Briefly, 5.0 g of $FeCl_2 \cdot 3H_2O$/$MnCl_2 \cdot 3H_2O$ was thoroughly dissolved in 300 mL of water and sonicated for 30 min. After achieving a homogenous solution, pre-treated biomass was mixed with the metal at a mass ratio of 1:1. The solution is magnetically stirred at 80 °C for 2 h and aged overnight. After soaking, the solution is centrifuged, washed with water until the pH is neutral, and oven-dried at 110 °C for 24 h. The dried sample was ground and passed through a 32-mesh sieve (0.50 mm) and pyrolyzed according to the procedure mentioned in the preparation of pristine biochar. The samples were labeled as SCBX-Fe and SCBX-Mn, where X denotes the optimum pyrolysis temperature.

### 3.4. Characterization of Biochar

The elemental composition of biomass prepared at different temperatures was detailed through CHN analysis by an elemental analyzer (EA, Elementar Analysensysteme GmbH Inc., Langenselbold, Germany). The surface morphology of the above-prepared catalytic biochar was characterized by scanning electron microscopy (SEM; Hitachi S-4700, Hitachi, Tokyo, Japan). The samples are mounted on a plate using carbon tape and coated with Platinum before SEM images were taken at different magnifications. Metal loading on biochar was measured using Energy Dispersive Spectroscopy (EDS, Horiba, 7200H, Tokyo, Japan) mounted on the SEM sample holder. The nitrogen adsorption/desorption isotherms, specific surface area, and pore structure (volume and size) were determined with a physisorption analyzer (Micromeritics Inc., ASAP 2020, Norcross, GA, USA). The $S_{BET}$ was obtained by the Brunauer−Emmett−Teller (BET) method, and the pore size distribution was plotted based on the Barrett−Joyner−Halenda (BJH) method from the desorption isotherm. All samples were oven-dried and degassed for 24 h at 115 °C before nitrogen adsorption to eliminate physisorbed moisture. The changes in functional groups before and after biochar modification were detected by a Fourier transform infrared spectrophotometer (FTIR, Perkin Elmer Inc., Spectrum 100, Waltham, MA, USA).

### 3.5. Evaluation of Ozone Removal Performance

Ozone adsorption experimental setup is portrayed in Figure S9. The catalytic performance of biochar for ozone elimination was performed in a fixed-bed flow quartz tube reactor (2 cm inner diameter) with a catalytic biochar loading of 0.03 g/cm$^3$ at different initial ozone concentrations for both pristine (5–10 ppm) and modified biochar (20–80 ppm). All gas supplies, including bypass lines, are controlled by flow controllers (Dwyer Company, RMA-26-SSV). Stable ozone flow was produced by an ozone generator (Lanyang Water Sample Technology Co., Ltd., 6g-O 3) and was diluted twice with air from an air compressor (FL air compressor, 1HP). An ozone detector (Henan Bosean Electronic Technology Co., Ltd., K-600, Zhengzhou, China) is placed at the discharge of the reactor responsible for measuring the changes in the concentration of ozone. The adsorption test is performed at atmospheric pressure and room temperature (23 °C). Diluted ozone is entering the reactor at a rate of 400 cc/min at 0.47 sec of retention time. For manipulating relative humidity, a silica gel column was placed at the inlet of both air and ozone gas and was monitored through an RH meter at the inlet of the reactor. The ozone removal performance of the catalytic biochar was reported through breakthrough curves as well as in terms of removal efficiency (Equation (6)) and adsorption capacity (Equation (7)).

$$Removal\ Efficiency(\%) = \frac{C_{in} - C_{out}}{C_{in}} \times 100 \tag{6}$$

$$Adsorption\ Capacity\ (mg/g) = \frac{\int_0^t (C_{in} - C_{out})\ Q\ dt}{W} \tag{7}$$

### 3.6. Adsorption Kinetics and Isotherm

The adsorption process and mechanism of ozone on catalytic biochar were revealed by kinetic simulations. Using the same breakthrough experiment, the ozone concentration at the outlet of the reactor was recorded every 20 min. Acquired kinetic data were modeled using pseudo first- order (Equation (8)), pseudo second- order (Equation (9)), intraparticle diffusion (Equation (10)) models:

$$\log(q_e - q_t) = \log q_e - k_1 t \tag{8}$$

$$\frac{t}{q_t} = \frac{1}{k_2 \cdot q_e{}^2} + \frac{1}{q_e}t \tag{9}$$

$$q_t = k_i t^{1/2} \tag{10}$$

where $k_1$, $k_2$, and $k_i$ are the pseudo-first-order, pseudo-second-order reaction and intraparticle diffusion rate constants, $q_e$ is the adsorption capacity at equilibrium (mg/g), and $q_t$ is the adsorption capacity (mg/g) at time $t$ (min).

Adsorption isotherm models were constructed to determine the maximum adsorption capacity of ozone on catalytic biochar. Isotherm data are generated by adjusting the initial ozone concentration, which ranges from 5–20 ppm for pristine biochar and 20–80 ppm for metal-impregnated biochar. Samples are contacted with ozone until saturation, and their isothermal data were modeled using the Langmuir (Equation (10)) and Freundlich model (Equation (11)).

$$q_e = \left( \frac{q_{max} \cdot K_L \cdot Co}{1 + K_L \cdot Co} \right) \tag{11}$$

$$q_e = K_F \cdot Co^{\frac{1}{n}} \tag{12}$$

where $K_L$ and $K_F$ are the Langmuir and Freundlich model constants (L/g), 1/n is a dimensionless exponent of non-linearity, $q_e$ is the adsorption capacity at equilibrium (mg/g), $q_{max}$ is the maximum adsorption capacity (mg/g) and $C_o$ stands for the initial ozone concentration (ppm).

## 4. Conclusions

This study pioneered demonstrating the ozone removal performance of biochar at ambient temperature. Due to increased human activity and rapid development across various industries, ground-level ozone has become an emerging pollutant that needs to be removed using more sustainable alternatives. The optimum pyrolysis of sugarcane-derived biochar was designated at 550 °C according to the biochar yield, surface chemistry, functional groups, surface morphology, and adsorption capacity. Catalytic biochar was successfully prepared by the facile incipient wet impregnation of Fe and Mn salts. Metal impregnation significantly improves the removal of ozone due to the enhanced morphological properties of biochar, such as surface area and pore volume. There is an approximately four-fold increase in the adsorption capacity of SCB550-Fe in contrast with pristine SCB550 during ozone adsorption at 20 ppm. SCB550-Fe demonstrated superior ozone removal performance compared to SCB550-Mn, exhibiting 122 mg/g capacity at 80 ppm. By virtue of isothermal and kinetic modeling, a plausible mechanism for ozone removal was proposed, including physical adsorption through electrostatic interaction and intraparticle diffusion, surface functional group complexation, and redox reactions. Future research should further elaborate on the validation of the mechanism of ozone and biochar interaction during the adsorption process as well as further modification to increase ozone removal efficiency compared to commercially available catalysts. This can be achieved by the analysis of compounds present in the exhaust gas of the reactor as well as further characterization of the used biochars, such as by X-ray photoelectron spectroscopy (XPS) analysis and density functional theory (DFT) calculation. Overall, due to availability, sustainability, and recent developments in biochar production and the lack of studies using biochar as an adsorbent for ozone removal, this study should provide preliminary insight into ozone removal through biochar and promote further research regarding material optimization and kinetic studies.

**Supplementary Materials:** The following supporting information can be downloaded at: https://www.mdpi.com/article/10.3390/catal13020388/s1, Figure S1: Pore sizes of pristine biochars pyrolyzed at 350 (**a**), 450 (**b**), 550 (**c**), and 650° (**d**).; Figure S2: Adsorption capacities of biochar prepared at different pyrolysis temperature.; Figure S3: EDS images of catalytic biochar SCB550-Fe and SCB550-Mn; Figure S4: Langmuir and Freundlich model fitting of pristine biochar SCB550 (**a**), and catalytic biochar SCB550-Fe (**b**) and SCB550-Mn (**c**) isothermal data; Figure S5: Pseudo-first (**a**) and pseudo-second (**b**) order kinetic modelling of SCB550.; Figure S6: Pseudo-first (**a**) and pseudo-second (**b**) order kinetic modelling of SCB550-Fe.; Figure S7: Pseudo-first (**a**) and pseudo-second (**b**) order kinetic modelling of SCB550-Mn.; Figure S8. Intraparticle diffusion modelling of SCB550-Fe.; Figure S9: Schematic diagram of ozone adsorption set-up utilizing an ozone generator (1), air compressor

(2), flowmeters (3), isothermal pressure tanks (4), quarts tube reactor (5), and ozone detector (6). and Table S1: Specific surface area comparison of previous studies with the results reported in this study.

**Author Contributions:** Conceptualization, R.A.V. and C.-T.C.; methodology, R.A.V. and C.-T.C.; software, R.A.V.; data acquisition and curation, R.A.V.; validation, R.A.V. and C.-T.C.; formal analysis, R.A.V.; investigation, R.A.V. and C.-T.C.; resources, C.-T.C.; data curation, R.A.V.; writing—original draft preparation, R.A.V.; writing—review and editing, C.-T.C. and A.R.C.; visualization, R.A.V.; supervision, C.-T.C. and A.R.C.; project administration, C.-T.C.; funding acquisition, C.-T.C. All authors have read and agreed to the published version of the manuscript.

**Funding:** This research was funded by Ministry of Science and Technology (MOST) Taiwan 109-221-E-197-017.

**Conflicts of Interest:** The authors declare no conflict of interest.

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
