# Peer review of "Facile Synthesis of Metal-Impregnated Sugarcane-Derived Catalytic Biochar for Ozone Removal at Ambient Temperature"

_catalysts, doi:10.3390/catal13020388_

Round 1

Reviewer 1 Report

Reviewer report on : “Facile Synthesis of Metal Impregnated Sugarcane-derived Cata-2 lytic Biochar for Ozone Removal at Ambient Temperature”, from Reginald A. Verdida,, Alvin R. Caparanga  and Chang-Tang Chang.

The manuscript studies a Fe and Mn containing  biochars to remove ground-level  ozone at ambient temperature. The biochar is produced from a sugarcane feedstock via co-pyrolysis of biomass and metal salts. A  mechanism for the  ozone removal by  biochar is proposed. The description of the different active sites regarding biochar/ozone interaction is complete and section 2.5 as well as the scheme in Fig. 5 represent a notable contribution regarding biochar and Metal+biochars materials acting as catalyst (although consideration detailed in point 12 should be considered). Besides, it is important to highlight that the samples are based on quite cheap materials.

The authors claim that their study constitutes a  preliminary insights into ozone removal using biochar based catalysts.

The work affords publication before the following points are addressed:

1-      Line 88, page 2, it is not clear if the authors refer to “biochar “ or “biomass”.

Finally, some studies also show that biochar with a lower ash content renders….”

2-      In page 3, “Recent studies have reported that transition metals (e.g., Fe, and Mn) introduced into carbon-based biomass precursors can augment the physicochemical properties of pristine biochar and lead to better catalytic performance [38].” Another publications regarding Fe-biochars could be cited. To label: Magnetic amendment material based on bio-char from edible oil industry waste. Its performance on aromatic pollutant removal from wáter, from Casoni et al. in Journal of Environmental Chemical Engineering, ournal of Environmental Chemical Engineering, Volume 8, Issue 2,  2020, 103559.  In that work results regarding the evidence for the presence of iron species (hematite, magnetite and maghemite) is shown.

3-      Page 5, lines 195-197. How can the authors conclude about nanometric scale pore size from SEM results?. Perhaps, the pore size data (1.58, 1.35, and 0.82nm) were taken from BET analysis (which are presented latter in the manuscript) ? Please, clarify on this.

4-      Regarding FTIR characterization and the corresponding discussion at the end of page 6. Firstly , it is not correct to name the IR bands as peaks. “Bands” and not “peaks” . Besides, Quantitative results, in order to be analyzed in FTIR spectra , should be considered by comparison between bands. I mean, it is not true that bands associated to functional groups, have decreased  in metal containing biochars. All the signal is lowered, as a whole. It is likely that a general lower transmittance of the sample is developed when metals are present. Conclusion about an increase in carbonization due to metal presence in not supported by FTIR results.

5-      Specific surface area, as calculated from BET analysis for pristine biochar is extremely high, by comparison with other reported results. Please, clarify on this. Please, carry out a comparison with already reported area results corresponding to biochars produced from sugarcane waste pyrolysis.

6-      Page 9: “Analogously, it can be observed that there’s an increase in average pore diameter of SCB550 (1.49 nm) compared to SCB550-Fe (1.76 nm) and SCB550-Mn (1.91 nm). This phenomenon might be due to adhesion of metal particles into the micropore structure of the  biochar, blocking the pores and reducing the pore diameter.”. If a pore blocking occurs , pore diameter would decrease and not increase.  The sentence is not clear at all. Please , rewrite.

7-      Page 9, line 302, for the biochar, is it correct to write “made”? It seems that the correct term is “tested”

8-      In page 9, “20ppm ozone level, SCB550 immediately reached equilibrium at 246 mins (Figure 5a)..”. What does “equilibrium” mean in this context? Thermodynamic equilibrium? Which is the species involved in such an equilibrium? Ozone is transformed into oxygen in a irreversible way, active sites are in some way saturated or poisoned, thus, it is not clear in which case equilibrium is present. Which is the equilibrium taking place in the surface scenario of ozone conversion to oxygen? Please clarify on this. The same for page 10, line 324.

9-      Regarding the  Kinetic Modelling in section 2.4, the main concern is that the isotherms under study both regards adsorption phenomena. No catalytic reaction is taken into account. Thus modelling is quite limited for this case. It is likely that the only valid conclusion is that Fe and Mn containing biochars are different from the pristine one, since the same model can not be applied for both cases. Please, comment on this.

A final consideration, related to point 7: for Freundlich and for Langmuir approximation, equilibra do take place

10-   Section 3.3. The amount of the Fe and Mn precursors solution and the corresponding concentration should be provided for the sake of reproducibility of sample preparation that other researchers could be interested in. In addition, metal loading (Fe and Mn wt%) should be reported in Table 2.

11-    Finally, a comment regarding the extent of possible irreversible adsorption of ozone by biochar (without subsequent conversion to oxygen) should appear . Is adsorption in all the cases followed by catalytic conversion to oxygen? Or certain amount of ozone remains bonded to biochar ( as ozone or as any other O species).?

Author Response

Response to the Reviewer 1:

1-      Line 88, page 2, it is not clear if the authors refer to “biochar“ or “biomass”.

Finally, some studies also show that biochar with a lower ash content renders higher porosity, and a higher ash content causes a lower surface area as ash blocks the pores of biochar, which ultimately confines its performance in most of its applications.

Response: Thank you for this insightful comment. From the literature survey, it appears that “biomass” is the more appropriate term to use since the topic of discussion is the pyrolysis of biochar precursors. This word located at line 88 is revised accordingly.

2-      In page 3, “Recent studies have reported that transition metals (e.g., Fe, and Mn) introduced into carbon-based biomass precursors can augment the physicochemical properties of pristine biochar and lead to better catalytic performance [38].” Another publications regarding Fe-biochars could be cited. To label: Magnetic amendment material based on bio-char from edible oil industry waste. Its performance on aromatic pollutant removal from wáter, from Casoni et al. in Journal of Environmental Chemical Engineering, ournal of Environmental Chemical Engineering, Volume 8, Issue 2,  2020, 103559.  In that work results regarding the evidence for the presence of iron species (hematite, magnetite and maghemite) is shown.

Response: Researchers of this study have gathered relatively good information from the abovementioned publication to support claims regarding enhancement on biochar properties upon metal modification. This article is therefore cited accordingly in the manuscript (reference number 42). This suggestion is much appreciated.

3-      Page 5, lines 195-197. How can the authors conclude about nanometric scale pore size from SEM results?. Perhaps, the pore size data (1.58, 1.35, and 0.82 nm) were taken from BET analysis (which are presented latter in the manuscript) ? Please, clarify on this.

Response: Thank you for asking this clarification. Pore sizes on SEM images of biochars pyrolyzed at different temperatures were measured and recorded (expressed in μm). Out of these representative pores, the average pore diameter is calculated respectively. Images are attached accordingly and are available in the appendix section (Figure A1) of the supplementary file.

4-      Regarding FTIR characterization and the corresponding discussion at the end of page 6. Firstly , it is not correct to name the IR bands as peaks. “Bands” and not “peaks” . Besides, Quantitative results, in order to be analyzed in FTIR spectra , should be considered by comparison between bands. I mean, it is not true that bands associated to functional groups, have decreased in metal containing biochars. All the signal is lowered, as a whole. It is likely that a general lower transmittance of the sample is developed when metals are present. Conclusion about an increase in carbonization due to metal presence in not supported by FTIR results.

Response: Thank you for your kind observation. All “peaks” used to describe the FTIR analysis are now changed to “bands”. The bands are indeed lowered in general and is assumed to be caused by the catalytic effects of metals during pyrolysis. Moreover, the increase in carbonization due to presence of metals can be properly observed by comparing SEM images of pristine biochar to Fe and Mn modified biochar where more pores and lattice defects are observed.

5-      Specific surface area, as calculated from BET analysis for pristine biochar is extremely high, by comparison with other reported results. Please, clarify on this. Please, carry out a comparison with already reported area results corresponding to biochars produced from sugarcane waste pyrolysis.

Response: A comparison with other reported results is such a good idea. Several previous studies have also reported approximately the same level of specific surface area as the ones demonstrated in this paper. These related studies are summarized in a table attached in the appendix section (Table A1) of the supplementary file.

6-      Page 9: “Analogously, it can be observed that there’s an increase in average pore diameter of SCB550 (1.49 nm) compared to SCB550-Fe (1.76 nm) and SCB550-Mn (1.91 nm). This phenomenon might be due to adhesion of metal particles into the micropore structure of the biochar, blocking the pores and reducing the pore diameter.”. If a pore blocking occurs, pore diameter would decrease and not increase.  The sentence is not clear at all. Please, rewrite.

Response: This comment is relatively correct. It is counter intuitive that the pore sizes of the biochars increase due to pore blocking. Perhaps the more correct explanation to the increase in pore sizes of metal impregnated biochars is the catalytic influence of Fe and Mn during the pyrolysis process. Metal salts have been found to further promote thermal decomposition of organic compounds during pyrolysis, which may have been the primary factor on the increase of pore size of the catalytic biochars. This statement is revised.

7-      Page 9, line 302, for the biochar, is it correct to write “made”? It seems that the correct term is “tested”

Response: The researchers agree that the more appropriate term to use is “tested”. Line 302 is revised. Thank you for your suggestion.

8-      In page 9, “20ppm ozone level, SCB550 immediately reached equilibrium at 246 mins (Figure 5a)..”. What does “equilibrium” mean in this context? Thermodynamic equilibrium? Which is the species involved in such an equilibrium? Ozone is transformed into oxygen in a irreversible way, active sites are in some way saturated or poisoned, thus, it is not clear in which case equilibrium is present. Which is the equilibrium taking place in the surface scenario of ozone conversion to oxygen? Please clarify on this. The same for page 10, line 324.

Response: Thank you for this comment. The equilibrium being discussed in this part refers to the saturation of active sites in the surface of the biochar with respect to ozone. The researchers determined that the more appropriate word to use is “saturation” to avoid confusion. This realization would not have been possible without your careful investigation.

9-      Regarding the Kinetic Modelling in section 2.4, the main concern is that the isotherms under study both regards adsorption phenomena. No catalytic reaction is taken into account. Thus modelling is quite limited for this case. It is likely that the only valid conclusion is that Fe and Mn containing biochars are different from the pristine one, since the same model can not be applied for both cases. Please, comment on this.

A final consideration, related to point 7: for Freundlich and for Langmuir approximation, equilibra do take place

Response: This was a completely valid insight. According to the kinetic modelling of this study, pseudo-second order modelling fits both Fe and Mn modified biochars ozone removal activity. This model suggests that the main mechanism of ozone removal is through chemisorption. This modelling has been the basis of the assumption that ozone reacts with the metals and surface oxygen-containing functional groups of the biochar. This phenomenon happens during reduction-oxidation reaction and surface complexation respectively which are both chemical reactions. Thank you for your thorough investigation.

10-   Section 3.3. The amount of the Fe and Mn precursors solution and the corresponding concentration should be provided for the sake of reproducibility of sample preparation that other researchers could be interested in. In addition, metal loading (Fe and Mn wt%) should be reported in Table 2.

Response: Thank you for this substantial input. The stoichiometric amounts of metal salts used during preparation of catalytic biochars is now indicated at the methodology section of the manuscript. As for the metal loadings, data are displayed in the EDS results shown in Figure 4e and 4f.

11-    Finally, a comment regarding the extent of possible irreversible adsorption of ozone by biochar (without subsequent conversion to oxygen) should appear. Is adsorption in all the cases followed by catalytic conversion to oxygen? Or certain amount of ozone remains bonded to biochar (as ozone or as any other O species).?

Response: The researchers would like to acknowledge this sensible comment. The extent to which ozone is being converted to other gaseous products or trapped in the pore structure of the biochar is one of the delimitations of this study which can be further investigated in future studies. This question renders an analysis of exhaust gas coming out of the fixed bed reactor to determine what compounds are produced. Also, further characterization such as XPS analysis of used biochar can help elucidate the conversion process. These recommendations are now indicated in the conclusion part of the manuscript (Page 16). Nevertheless, the researchers hope for your understanding as this presents a first study paper and aims to provide preliminary insights regarding ozone removal using catalytic biochar.

Reviewer 2 Report

1. Introduction: Should address more on the recent advances about the biochar-supported Fe/Mn minerals and their applications in pollution control.

2. Abstract: The conclusion of optimum pyrolysis temperature of 550 C must be supported with more solid evidence. It is strongly recommended to compare the performance of biochar-supported Fe/Mn minerals prepared at other temperatures (e.g. 650 C) on ozone removal.

3. Line 195: The discussion about the porosity of biochar using SEM observation is questionable. SEM cannot give the pore structure at nanometer scale.

4. More information about the Fe and Mn species in the modified biochar must be provided, as such information is crucial for understanding the mechanisms of ozone removal.

5. The discussion about the ozone removal mechanisms must be supported by solid evidence, for example, the change of modified biochar after ozone removal, and the products obtained from ozone decomposition.

Author Response

Response to Reviewer 2:

  1. Introduction: Should address more on the recent advances about the biochar-supported Fe/Mn minerals and their applications in pollution control.

Response: This is a relatively good suggestion that the researchers have missed. Related studies about Fe/Mn modified biochar are now included in the manuscript. Thank you for this suggestion.

  1. Abstract: The conclusion of optimum pyrolysis temperature of 550 C must be supported with more solid evidence. It is strongly recommended to compare the performance of biochar-supported Fe/Mn minerals prepared at other temperatures (e.g. 650 C) on ozone removal.

Response: The researchers appreciate this concern. Concluding that the optimum pyrolysis temperature as 550°C was reasonably supported with enough characterization analysis. Surface chemistry, surface functional groups and surface morphology of biochars pyrolyzed at various temperatures are also thoroughly investigated and elucidated in order to establish this conclusion.

  1. Line 195: The discussion about the porosity of biochar using SEM observation is questionable. SEM cannot give the pore structure at nanometer scale.

Response: Thank you for your careful investigation. The average pore size of pristine biochars is recorded using SEM images in the micro scale. These SEM images are now attached in the appendix section (Figure A1) of the manuscript accordingly for reference.

  1. More information about the Fe and Mn species in the modified biochar must be provided, as such information is crucial for understanding the mechanisms of ozone removal.

Response: This is a completely valid insight. The amount of Fe and Mn loadings on the biochar as well as its degree of dispersion have been provided using EDS elemental mapping. Another beneficial characterization would be an XPS analysis which is one of the delimitations of this study. This can be a good starting point for future studies in order to fully elucidate the mechanism of ozone removal. Thank you for your suggestions.

  1. The discussion about the ozone removal mechanisms must be supported by solid evidence, for example, the change of modified biochar after ozone removal, and the products obtained from ozone decomposition.

Response: The researchers would like to acknowledge this excellent insight. Proposed mechanisms of ozone removal are determined by rigorous rection analysis of both ozone molecule and biochar surface chemistry based on their existing physiochemical characteristics. Ozone is a permanent dipole molecule allowing for a great possibility of electrostatic interaction with the surface of the biochar. Additionally, ozone is a known oxidizing agent that can easily interact with the metal particles disperse in the surface of catalytic biochar. Furthermore, related studies that previously explain the ozone removal or decomposition were used to support these proposed mechanisms. Finally, recommendations regarding further characterizations for future studies are presented in the conclusion part of the manuscript (Page 9) in order to support the proposed mechanism. Nevertheless, the researchers hope for your understanding as this presents a first study paper and aims to provide preliminary insights regarding ozone removal using catalytic biochar.

Round 2

Reviewer 1 Report

The manuscript is publishable in the present form. The authors have notably improved the manuscript. The signifcance of content is high.  

Author Response

Response to Reviewer 1 (Round 2)

The manuscript is publishable in the present form. The authors have notably improved the manuscript. The significance of content is high. 

Response: Thank you for your keen insights and suggestions.

Reviewer 2 Report

The manuscript must be improved following the reviewer's advice provided in the first round review, and give  convincing reasons for why they did not do so.

Author Response

Response to Reviewer 2 (Round 2)

The manuscript must be improved following the reviewer's advice provided in the first round review, and give convincing reasons for why they did not do so.

  1. Introduction: Should address more on the recent advances about the biochar-supported Fe/Mn minerals and their applications in pollution control.

Response: The introduction part of the manuscript is now reinforced with related literature regarding recent advancements in Fe/Mn supported biochar for different applications. This can be seen on Page 3, Lines 109 to 114. Additionally, a summary table of this literature has been prepared.

Biochar Name

Application

Pollutant

Reference

Fe2O3/Fe3O4/WBC

wastewater remediation

Cr (IV) ions

[39]

Fe-N-C-x

wastewater remediation

PMS activation for organic pollutant

[41]

MBZX

wastewater remediation

guaiacol

[42]

MFSCBB-MCs

wastewater remediation

lead

[43]

  1. Abstract: The conclusion of optimum pyrolysis temperature of 550 C must be supported with more solid evidence. It is strongly recommended to compare the performance of biochar-supported Fe/Mn minerals prepared at other temperatures (e.g. 650 C) on ozone removal.

Response: The conclusion of optimum pyrolysis temperature of 550°C is supported by the and characteristics analysis results from Elemental Analysis (EA), FTIR spectra, and SEM images of biochar at different temperatures as well as adsorption capacity assessment. EA of the biochar shows a decrease in O content and O/C ratio as pyrolysis temperature increases. This translates to a decrease in polarity and oxygen-containing groups of the biochar which is not good for better removal of oxygen. This means pyrolysis temperature of 650°C is not an efficient temperature. These results are provided in Pages 3 to 4, Lines 148 to 153. FTIR analysis provided in page 4, lines 162 to 181 also clearly demonstrates the decrease in functional groups of the biochar especially oxygen-containing groups. Again, these groups are deemed to be beneficial for ozone removal. This means pyrolysis temperature of 650°C is not an efficient temperature. SEM images provided at Figure 2 (Page 6) shows a more defined pore structure as the pyrolysis temperature increases. The biochar pyrolyzed at 350°C and 450°C will not be a good candidate since the pores are not well defined. Finally, pore sizes from SEM images provided in Figure A1 show that the biochar produced at 550°C has the lowest average pore size among the 4 samples. Small pore size is a very good property of biochar beneficial for good ozone removal.

            Additionally, performance of biochar pyrolyzed at 350, 450, 550, and 650°C on removal of ozone is evaluated at 15ppm. It was found that among the biochar samples, the highest adsorption capacity recorded belongs to biochar pyrolyzed at 550°C at 8.23 mg/g. These results are now indicated in the manuscript on page 6 in lines 222 to 226. Adsorption capacities of all biochar prepared at different pyrolysis are presented in Figure A2 in the appendix part of the supplementary file.

  1. Line 195: The discussion about the porosity of biochar using SEM observation is questionable. SEM cannot give the pore structure at nanometer scale.

Response: Page 5 Lines 199 and 200 are now corrected to clarify that the pore sizes recorded from SEM images are measured in the microscale, not in nanometer scale. The SEM images containing these measurements are provided in Figure A1 at appendix part of the supplementary file. Additionally, average pore size of the Fe and Mn modified biochar are provided using BET calculations (nanoscale).

  1. More information about the Fe and Mn species in the modified biochar must be provided, as such information is crucial for understanding the mechanisms of ozone removal.

Response: Information about Fe and Mn species impregnated on biochar are provided using Energy-dispersive spectroscopy (EDS) analysis. Figure 4.2 and Figure 4.f (Page 8) shows very minimal content of Fe and Mn loading on surface of the biochar (only 4.5% and 2.1% weight, respectively). Additionally, EDS elemental mapping provided in Figure A2 in the appendix part of the supplementary file shows that Fe and Mn species are relatively dispersed evenly in the surface of the biochar.

As for the mechanism of ozone removal by Fe and Mn, the manuscript has been revised on page 13 in lines 409 - 418 to cite some relevant studies explaining the decomposition of ozone by Fe and Mn metals. Most of them suggest that ozone is decomposed by metals through oxidation-reduction reaction as presented in Equations 3 to 5 on page 13 of the manuscript. Hereunder, a summary table about these related studies has been provided.

Material Name

Remarks

Reference

X@PB (X= Fe, Mn, Zn)

Existing transition metal oxides enhance redo-active active sites of biomass allowing for enhanced electron transfer capabilities.

[77]

Mn/Al2O3

Elucidated the reaction mechanism of ozone decomposition by Mn metals

[78]

MnOx/biochar and FeOx/biochar

Elucidated the reaction mechanism of ozone decomposition by Fe and Mn metals.

[79]

  1. The discussion about the ozone removal mechanisms must be supported by solid evidence, for example, the change of modified biochar after ozone removal, and the products obtained from ozone decomposition.

Response: FTIR of the fresh and used sample and intraparticle diffusion modelling have been performed to further provide solid evidence to support the proposed ozone removal mechanism. Chemisorption is now supported by an FTIR analysis of fresh and used biochar provided on page 13 in Figure 7 of the manuscript. The results are also indicated on page 12 in lines 400 to 405 to support the proposed ozone removal mechanism through chemisorption. It can be observed that surface functional groups and particularly oxygen-containing groups were significantly decreased after adsorption of ozone.

Figure 7. Comparison of FTIR spectra of fresh and used SCB550-Fe. (Please see the attachment)

Physical adsorption by pore diffusion is now supported by intraparticle diffusion modelling. Summary of parameters calculated from the modelling is provided in Table 6 on page 13. It is indicated that the physical adsorption of ozone through bulk diffusion, film diffusion and pore diffusion are proven through intraparticle diffusion modelling with 3 stages of adsorption on page 12 in lines 398 to 400, as shown in Figure A8.

Figure A8. Intraparticle diffusion modeling of SCB550-Fe (Please see the  attachment)

Oxidation and reduction mechanisms of ozone removal are supported by previous studies that elucidated the stages of how ozone is being decomposed by metals such as Fe and Mn. These references are cited on page 9 in lines 409 to 418 of the manuscript.
